# A Green Supply Chain Member Selection Method Considering Green Innovation Capability in a Hesitant Fuzzy Environment

Jiafu Su [1,2] , Baojian Xu [1], Lvcheng Li [3,*], Dan Wang [1] and Fengting Zhang [1]

1   National Research Base of Intelligent Manufacturing Service, Chongqing Technology and Business University, Chongqing 400067, China
2   International College, Krirk University, Bangkok 10220, Thailand
3   School of Entrepreneurship, Wuhan University of Technology, Wuhan 430070, China
*   Correspondence: lvcheng.li@whut.edu.cn

**Abstract:** The purpose of this paper is to propose an improved hesitation fuzzy multi-attribute decision-making method to realize green supply chain member selection under green innovation vision. The method uses hesitation fuzzy sets to express decision information of decision makers, takes green innovation capability as the evaluation perspective, and selects green innovation input, synergy of subjects in green supply chain, green innovation output capability, institutional innovation capability of enterprises in green supply chain, and green innovation sustainability as the indexes to evaluate the green innovation capability of enterprises. The multi-attribute decision method proposed in this paper takes into account the shortcomings of the original hesitant fuzzy multi-attribute decision method considering attribute weight optimization in the determination of attribute weights and scheme ranking, then proposes a three-point estimation method for scheme ranking and optimizes the attribute weights by quantifying the balance coefficients of the original decision method. Finally, an example is used to verify the rationality and effectiveness of the proposed method, and a comparison with the original method is made to highlight the advantages of this paper's method. This paper provides a certain theoretical basis for the selection of members in green supply chains, which helps the selection of members in green supply chains and provides some insight for similar hesitant fuzzy multi-attribute decision-making problems in other fields. In future research, the method proposed in this paper can be considered to combine with probabilistic hesitant fuzzy sets and some other fuzzy sets for method extensions to solve multi-attribute decision-making problems.

**Keywords:** green innovation capability; green supply chain; attribute weight optimization; hesitant fuzzy multi-attribute decision making; three-point estimation method

**MSC:** 03E72

## 1. Introduction

In order to deal with global climate change and environmental pollution, governments around the world have proposed carbon peaking and carbon neutrality strategic tasks to reduce resource consumption and promote green and low-carbon development [1]. For example, China has proposed policies for 2021 that will address significant challenges in green technology innovation, speed up the shift to environmentally friendly development, and establish a market-driven green technology innovation system. In recent years, the U.S. government has also been promoting the "Green New Deal", which aims to combine renewable energy, resource efficiency, a low-carbon economy, and other green ideas with its economic policies to address climate change, develop a low-carbon economy, and pursue environmental and social justice [2]. In 2020, the Japanese government released the "Green Growth Strategy for Achieving Carbon Neutrality by 2050", which aims to promote the green development of Japan's low-carbon industries by providing policy support in three areas: fiscal, taxation, and finance [3]. Nowadays, a large number of companies are

committed to the development and production of green products, and more and more consumers are considering environmental factors when purchasing goods or services [4]. In order to achieve the least impact on the environment and the highest resource efficiency, the Manufacturing Research Society of Michigan State University in the United States first proposed the concept of the green supply chain based on green manufacturing theory and supply chain management technology [5], and the green supply chain has been widely recognized for its good balance of environmental factors and social benefits in supply chain management. At the same time, the green supply chain is a subsystem of the ecological economic system [6], which makes the development of the green supply chain usher in an opportunity.

As a modern management model based on green manufacturing theory and supply chain management technology, the green supply chain integrates environmental impact and resource efficiency in the entire supply chain and green design of the entire supply chain from the product life cycle, involving suppliers, manufacturers, distributors, retailers and other enterprises, and end users. In addition, in order to accomplish coordination and optimization of economic and social benefits, each firm in the supply chain and its internal departments must work closely together [7]. However, in the green supply chain, suppliers play a key role in the supply of green raw materials and production of green products, which determines the supply source of the green supply chain and is crucial to the smooth implementation of the green supply chain [8]. Similarly, for manufacturers, they directly decide the green production of products. For sellers, they directly decide the green sales of products. For users, they directly decide the green use of products. All these processes are also crucial to the smooth implementation of the green supply chain. Whether it is suppliers, manufacturers, distributors, retailers, or end-users, there is a mutual choice between them. For the whole supply chain, whether it is a matter of suppliers choosing manufacturers, or manufacturers choosing suppliers, or manufacturers choosing distributors, or distributors choosing retailers, it is a matter of member selection in the supply chain. Moreover, the trust relationship among members in the supply chain affects the mutual cooperation among members [9]. Therefore, it is necessary to study a method that is applicable to the selection of member relationship in the green supply chain.

Unlike traditional supply chains, green supply chains need to take into consideration the triple benefits of economy, environment, and society [10]. Therefore, in the selection of green supply chain members, the economic, environmental, and social benefits should also be integrated into the selection process. Chen et al. [11] proposed green innovation and defined it as the creative activity of human society that focuses on the harmonious development of environment-economy-society and makes it possible to be called green innovation. From this definition, it is easy to find that green innovation is a good balance of economic, environmental, and social benefits. Further, the above problems have been transformed into the selection of green supply chain members from the perspective of green innovation. Some scholars have found that green innovation cooperation in the supply chain is an important way to strengthen the core technology capability of enterprises, optimize the allocation of internal and external innovation resources, and enhance the stability and reliability of external cooperation strategies [12]. Therefore, this paper chooses to study the selection of green supply chain members based on the perspective of green innovation capability. This study offers an original contribution to the research on the subject. In the selection of green supply chain members, the previous literature mainly focused on the selection of manufacturers and suppliers. In the selection method, qualitative methods were mainly used for selection evaluation. In the evaluation indicators, the selection of indicators is broad but not in-depth. In this paper, when researching the selection of members of the green supply chain, we select green innovation capability as the criterion for evaluating members, and the selection of members is not limited to suppliers and manufacturers but involves companies in the whole supply chain. At the same time, the selection of members in the green supply chain based on green innovation capability will have many advantages. For example, for enterprises, it can improve their product

innovation ability, meet the green demand of consumers, and meet the needs of social development, which is beneficial to their own development. At the same time, for the whole green supply chain, it can enhance the green innovation ability of the whole green supply chain and make favorable contribution to a series of environmental problems such as climate change, carbon neutral, and carbon peak.

When evaluating the selection of green supply chain members from the perspective of their capacity for green innovation, it is taken into account that because decision-makers have different ways of thinking and knowledge, they might assign different decision values to the same decision. As a result, the scoring will be ambiguous, and the selection evaluation results will vary as a result of the various ways in which the weights of each index have been determined. Hesitant fuzzy sets, as a kind of fuzzy set, can well solve the fuzziness and hesitancy of experts at the subjective level. Among them, fuzziness refers to the fact that, many times, experts cannot give a specific assessment value when evaluating objectives, but can only give a vague and unclear range; hesitancy refers to the fact that when experts evaluate objectives, the assessment value will hesitate between several possible values [13]. With the development of economy and society, realistic decision-making problems are becoming more and more complex, and people often need to evaluate and make decisions on a problem from multiple aspects and dimensions, and the ambiguity and hesitancy in evaluating the problem are becoming stronger. In supply chains, to determine which supplier is the best, various fuzzy methods have been used by combining economic and ecological criteria [14]. Therefore, this paper proposes an improved hesitant fuzzy multi-attribute decision-making method. This method chooses the hesitation fuzzy theory as its theoretical underpinning and is based on the improved hesitation fuzzy multi-attribute decision-making method taking attribute weight optimization into consideration for selection decision. In this method, decision making can fully utilize the decision information provided by each decision maker and give each attribute scientific and reasonable weights, while also combining risk-optimism to rank the solutions.

In summary, this paper proposes an improved hesitant fuzzy multi-attribute decision method for member selection in green supply chains, which is applicable to the inter-firm selection decision problem in green supply chains. In addition, this paper has the following contributions:

Firstly, this paper improves the hesitant fuzzy multi-attribute decision-making method considering attribute weight optimization and provides a more scientific and reasonable attribute weight determination method and a brand-new decision result sorting method for the hesitant fuzzy multi-attribute decision-making problem.

Secondly, the selection of green supply chain members based on the perspective of green innovation in this paper can promote the innovative research and development of green products by enterprises, then enhance the green innovation ability of the whole green supply chain, and, at the same time, provide ideas for future related studies and theoretical references for enterprises when selecting green supply chain members.

Finally, in this paper, the research on the selection of members in green supply chains is no longer limited to a certain two subjects, but the entire supply chain is studied when the selection decision is made among the enterprises. Therefore, this paper provides a more applicable green supply chain member selection method.

The rest of the paper is as follows. Section 2 reviews the current literature on green supply chain membership selection and related selection methods and evaluation metrics. Section 3 presents the theoretical basis of the research methods involved in this paper. Section 4 gives a relevant description of the research problem of this paper. Section 5 gives the decision method of this paper. Section 6 shows the application of the method to an example. Section 7 compares the research methodology of this paper with that of the previous literature based on the numerical examples. Section 8 draws the conclusions of this paper.



## 2. Literature Review

In recent years, scholars have focused their research on green supply chains on the distribution of benefits among supply chain members, operation optimization, cost management, product pricing, and environmental impact, while research on the selection of green supply chain members has mainly focused on the selection of suppliers. Xiao et al. [8] studied the supplier selection and coordination strategy of B2C enterprises, in addition to the consideration of the supplier's green concept in the selection of suppliers, and also a comprehensive assessment of the supplier's reputation, corporate equipment, production capacity, technology level, innovation, after-sales service, product price, environmental awareness, re-production capacity, etc., and through the construction of a coordination model to identify suppliers. By applying the Pythagorean cubic fuzzy Hamacher aggregation operator to the green supply supplier selection problem, Abdullah et al. [15] found that when using this method for green supply chain supplier selection, they can find the best green suppliers for green supply chain management better than traditional methods. Cimyari et al. [16], through the case study of the Sazeh Gostar Saipa photovoltaic company, active in the photovoltaic industry, studied the issue of selecting green suppliers from three aspects of economy, environment, and social responsibility under uncertain circumstances. In summary, scholars have conducted supplier selection mainly in terms of economic, environmental, and social responsibility. However, green innovation capabilities can be well integrated into these three aspects [17]. Therefore, this paper will conduct the selection of green supply chain members from the perspective of green innovation.

Innovation, as a direct way to improve the competitiveness of enterprises, regions, and countries at all levels, is the main driver of economic growth [18]. Firms are now under constant pressure from various governmental and non-governmental agencies to shift from traditional environmentally polluting products to green product innovation (GPI) [19]. This makes green innovation capability especially critical in green supply chains, and, at the same time, how to scientifically and reasonably evaluate the green innovation capability of supply chain members has become an important research issue in this paper. At present, scholars have conducted a large number of studies on the evaluation of innovation capability, but there are some differences in the use of evaluation indexes. Su et al. [20] selected four indicators of R&D capability, financial capability, synergy capability, and output capability as evaluation indicators in the evaluation of enterprise technology innovation capability. Li et al. [21] constructed an indicator system with the core of main body synergy, innovation input, innovation environment, and innovation output for the evaluation of enterprise technology innovation capability. Musaad et al. [22] selected four green innovation criteria for green supplier selection from existing studies: green innovation capacity, green innovation initiatives, green innovation performance, and green innovation monitoring and follow-up. Based on the above-mentioned scholars' research on innovation capability evaluation indexes, this paper constructs five indexes to evaluate the green innovation capability of enterprises: green innovation input, synergy of subjects in the green supply chain, green innovation output capability, institutional innovation capability of enterprises in the green supply chain, and green innovation sustainability.

However, the above literature does not take into account the fact that the same decision indicator will be scored by different decision makers in the partner selection process. Thus, when different decision makers score each indicator, there will be a situation where one indicator is associated with multiple scoring values, i.e., different decision makers will give different decision values under one attribute. In addition, the research method in the above-mentioned literature only determines the indicators and then makes partner selection directly on the basis of the indicators. In view of the shortcomings in the above literature, this paper makes selection decisions based on hesitant fuzzy sets theory when selecting members of the green supply chain. Yazdani et al. [23] propose a two-stage integrated decision model using fuzzy TOPSIS and LP in an attempt to develop a mathematical model that can be applied to solve the combined supplier selection and order allocation problem. For the hesitant fuzzy multi-attribute decision-making methods, Krishankumar et al. [24]

considered that hesitant fuzzy linguistic term sets (HFLTS) cannot be used to represent complex linguistic terms. To better circumvent this problem, a dual-level HFLTS (DHH-FLTS) is proposed, and a decision framework is proposed in the context of DHHFLTS. Peng et al. [25] derived a hesitant fuzzy group decision-making method with multiple biases by incorporating multiple bias information into the aggregation of decision information via the multidimensional induced ordered weighted average (MIOWA) operator. Li et al. [26] proposed the hesitant fuzzy variable-weight integrated decision model under the consideration of the influence of the equilibrium between the state values of the factors on the factor weights, derived the variable-weight integrated decision method in the hesitant fuzzy environment, and gave its application to the group decision cluster. Considering the advantages of hesitation fuzzy sets in expressing hesitation fuzzy information and the advantages of multi-attribute decision making, this paper will describe each decision maker's evaluation of the same attribute by hesitation fuzzy sets and form hesitation fuzzy elements. Finally, all hesitation fuzzy elements containing decision-maker information are formed into a hesitation fuzzy decision matrix for decision making to obtain a hesitation fuzzy multi-attribute decision-making method.

The aforementioned literature tends to prefer these decisions as a simple one when making decisions; however, in reality, the selection of members in a supply chain is a complex choice process that requires the evaluation of possible courses of action or options by selecting a preferred option or ranking the options from best to worst, which is a process of multi-attribute decision analysis. In daily practice, the use of MADA is essential to signal the best rational choice to the decision maker so that he can allocate limited resources among competing and alternative interests [27]. In the multi-attribute decision problem, two factors, attribute weights and attribute values, directly affect the decision results [28]. At present, many scholars have given many decision methods for the attribute weight problem in the decision problem under hesitant fuzzy multi-attribute. Liu et al. [29] proposed a hesitant fuzzy multi-attribute decision method considering attribute weight optimization in order to synthesize the decision information of decision makers. Cheng et al. [30] proposed a weight-optimized improved TODIM decision method for the hesitant fuzzy multi-attribute decision problem with unknown attribute weights. Starting from two levels of solution and attribute values, a multi-objective optimization model is constructed with hesitant fuzzy number mean, variance, and non-explicit entropy to determine the relative weights. Zhang et al. [31] proposed an objective weighting approach based on Shannon information entropy, which expresses the relative intensities of attribute importance to signify the average intrinsic information transmitted to the decision maker. Furthermore, they construct a hesitant fuzzy MADM approach based on the TOPSIS method and a weighted correlation coefficient proposed in their paper. The above-mentioned studies use quantitative methods to optimize the attribute weights, and then combine those weights with decision-making methods to apply the optimized weights to decision-making problems. As a result, they avoid the shortcomings of subjective weight assignment, which is heavily dependent on the decision maker due to its subjectivity. Based on this, this paper proposes an improved hesitant fuzzy multi-attribute decision making method considering attribute weight optimization. The research method proposed in this paper quantifies each balance coefficient according to the nature of balance coefficients based on the research in the literature [29] and further enhances the scientificity and rationality of decision making by introducing the three-point estimation method to rank the final decision scheme.

In summary, by processing the equilibrium coefficient method and solution ranking method in the decision-making method based on the literature [29], this paper proposes an improved hesitant fuzzy multi-attribute decision-making method for green supply chain member selection taking attribute weight optimization under the perspective of green innovation capability. The method uses hesitation fuzzy sets to express decision information of decision makers and selects green innovation input, synergy of subjects in green supply chain, green innovation output capability, and institutional innovation

capability of enterprises in green supply chain and sustainability of green innovation as evaluation indexes. The specific steps of the method can be found in Section 5.2.

## 3. Preliminaries

### 3.1. Hesitation Fuzzy Sets Theory

In a realistic green supply chain member selection, decision makers always face various uncertainties, which make them hesitate to evaluate the selection of decision objects, thus making it difficult to obtain a decision result that combines information from all decision makers. Based on this situation, some scholars have proposed the hesitant fuzzy sets theory, and the following will briefly introduce the definition of hesitant fuzzy sets theory.

**Definition 1** ([32,33]). *Let X be a fixed set, then the hesitation set is a function that maps each element of X to a subset of [0, 1].*

To facilitate understanding, Xia and Xu [34] first expressed hesitant fuzzy sets in mathematical form: $W = \{< \mu, h_w(\mu) > | \mu \in X\}$, whose $h_w(\mu)$ denotes some possible affiliations of the element $\mu$ with respect to the set $W$, and $h_w(\mu)$ is the set of some numbers in [0, 1], and called $h = h_w(\mu)$ a hesitant fuzzy element.

When estimating the degree to which the solution $H$ satisfies the attribute $M$, it is described by the hesitation fuzzy element since it cannot be described specifically by a specific value. However, it is often necessary to compare hesitant fuzzy elements when performing scheme comparison, so Xia and Xu [34] gave the score function: $s(h) = \frac{1}{l_h} \sum_{a \in h} a$, where $h$ is the hesitant fuzzy element and $l_h$ is the number of elements in $h$, which is based on the score function ranking, i.e., for any two hesitant fuzzy elements $h_1, h_2$, if $s(h_1) < s(h_2)$, then $h_1 < h_2$; if $s(h_1) > s(h_2)$, then $h_1 > h_2$; if $s(h_1) = s(h_2)$, then $h_1 \sim h_2$. Some scholars define the deviation degree [35]: $v(h) = \frac{1}{l_h} \sqrt{\sum_{\gamma_i, \gamma_j \in h} (\gamma_i - \gamma_j)^2}$, where $\gamma$ is the element in $h$, based on the deviation degree and score function for ranking of hesitant fuzzy elements.

**Definition 2** ([32]). *Let the score function and variance function of the hesitant fuzzy element $h_1(x)$ be $s(h_1)$, $v(h_1)$, and the score function and variance function of $h_2(x)$ be $s(h_2)$, $v(h_2)$, respectively, then:*

(1) *If $s(h_1) < s(h_2)$, then $h_1 < h_2$*
(2) *If $s(h_1) = s(h_2)$, then:*

  *If $v(h_1) = v(h_2)$, then $h_1 = h_2$;*
  *If $v(h_1) > v(h_2)$, then $h_1 < h_2$;*
  *If $v(h_1) < v(h_2)$, then $h_1 > h_2$.*

**Definition 3** ([36]). *Let the hesitant fuzzy element $h(y) = \{\vartheta_j | j = 1, 2, \ldots, l_h\}$, where $h^+$ and $h^-$ denote the maximum and minimum values in $h(y)$, respectively, i.e.,*

$$h^+ = \max\{\vartheta_j | j = 1, 2, \ldots, l_h\}$$

$$h^- = \min\{\vartheta_j | j = 1, 2, \ldots, l_h\}$$

### 3.2. Three-Point Estimation Method

The three-point estimation method is a calculation method for calculating construction durations that originated from the Program Evaluation and Review Technique (PERT), also known as the three-time estimation method in the field of engineering construction. The method takes into account the complexity and uncertainty of actual project management, which makes it difficult to estimate the duration of a project activity in project management, and, therefore, estimates the duration of a project by using optimistic time ($t_o$), pessimistic

time ($t_p$), and the most likely time ($t_m$) to calculate the expected duration of a certain activity ($t_e$) [37]. Its calculation formula is as follows.

$$t_e = \frac{t_o + 4t_m + t_p}{6}$$

where $t_o$ indicates the time to complete the project activity without any negative impacts, i.e., optimistic time. $t_p$ indicates the time to complete the project activity under all negative impacts, i.e., pessimistic time. $t_m$ denotes the time to complete the project activity under normal conditions, i.e., the most likely time.

As there are three types of risk-optimisms for decision makers in hesitant fuzzy decision making (risk-pessimism, risk-neutral, and risk-optimism), the decision-making process is as complex and uncertain as determining the project activity time. Therefore, in order to take into account the scientific and rational nature of the decision results and the fact that, in some cases, the decision maker cannot discern which of the three risk preferences he or she belongs to, this paper proposes a three-point estimation method decision based on the above method, i.e., ranking the solutions according to their expected similarity ($S_e(H_i)$) to the positive ideal point ($H^+ = \{h_1^+, h_2^+, h_3^+, \ldots\ldots, h_m^+\}$). We designate the scenario by H, and h denotes a set of evaluations of the situation H made by the decision maker, i.e., hesitant fuzzy elements. The decision formula is:

$$S_e(H_i) = \frac{S_o(H_i) + 4S_m(H_i) + S_p(H_i)}{6} \tag{1}$$

$$S(H_i) = \left[1 - d_\mu\left(H_i, H^+\right)\right] \tag{2}$$

$$d_\mu\left(H_i, H^+\right) = \sum_j^m \mu_j \sqrt{\frac{1}{l_j} \sum_{k=1}^{l_j} \left|h_{ij}{}^{\tau(k)} - (h_j^+)^{\tau(k)}\right|^2} \tag{3}$$

where $S_e(H_i)$ denotes the expectation similarity of the $i$-th scheme $H_i$, $S_p(H_i)$ denotes the risk-pessimism similarity of the $i$-th scheme $H_i$, $S_m(H_i)$ is the risk-neutral similarity of the $i$-th scheme $H_i$, and $S_o(H_i)$ denotes the risk-optimism similarity of the $i$-th scheme $H_i$. Additionally, the larger $S_e(H_i)$ is, the better the scheme is. $d_\mu(H_i, H^+)$ denotes the hesitant Euclidean weighted distance of the solution from the positive ideal point. When the number of elements in $h_{ij}$ is less than that in $h_j^+$, the similarity is added according to the required calculation. When calculating $S_p(H_i)$, $\beta = h^-$ is added according to the risk-optimism type. When calculating $S_o(H_i)$, $\beta = h^+$ is added according to the risk-pessimism type. When calculating $S_m(H_i)$, $\beta = \frac{h^- + 4\hat{h} + h^+}{6}$ is added according to the risk-neutral type, where $\hat{h}$ is the average of the remaining elements after dropping $h^-$ and $h^+$ ($\hat{h}$ is zero when $l_h < 3$). $\tau(k)$ denotes the $k$-th smallest element in $h_{ij}$.

## 4. Problem Description

This study will address the problem of green supply chain member selection by decision makers in the perspective of green innovation. Suppose the decision maker evaluates $n$ members under $m$ attributes by the decision-making method proposed in this paper and outputs the evaluation results as a decision matrix. The similarity of each scenario in the risk-optimistic, risk-neutral, and risk-pessimistic scenarios is then calculated based on the attribute weights and the decision matrix, using the formula for obtaining the attribute weights following parameter optimization provided in Section 5.1 of this paper. Finally, the alternatives are ranked according to Equations (1)–(3), and the green supply chain member that meets the decision maker's needs is selected. To facilitate the calculation, the parameters are specified in this paper as follows:

$H = \{H_1, H_2, H_3, \ldots, H_n\}$: a set of green supply chain member selection options based on green innovation capability, where $n \in N^+$, $H_i$ denotes the $i$-th green supply chain member selection option, $i = 1, 2, 3, 4 \ldots \ldots n$.

$M = \{M_1, M_2, M_3, \ldots \ldots M_m\}$: a set of attributes of the green supply chain member selection scheme based on green innovation capability, where $m \in N^+$, $M_j$ denotes the $j$-th attribute, $j = 1, 2, 3, 4 \ldots \ldots m$.

$\mu = \{\mu_1, \mu_2, \mu_3, \ldots, \mu_m\}$: a set of weights for each attribute, where $m \in N^+$, $\mu_t \in [0, 1]$, $\sum\limits_{t=1}^{m} \mu_t = 1$.

$D = \{h_{ij}\}$: a green supply chain member selection hesitation fuzzy decision matrix, where $h_{ij}$ $(i = 1, 2, 3, \ldots, n, \ j = 1, 2, 3, \ldots, m)$ is a hesitation fuzzy element of scheme $H_i$ satisfying attribute $M_j$.

Let the vector form of the scheme $H_i (i = 1, 2, 3, \ldots \ldots n)$ be $H_i = \{h_{i1}, h_{i2}, h_{i3}, \ldots \ldots, h_{im}\}$.

In this paper, the positive ideal point is used as the decision point, i.e., the hesitant fuzzy positive ideal point is assumed to be $H^+ = \{h_1^+, h_2^+, h_3^+, \ldots \ldots, h_m^+\}$, where:

$$
h_j^+ = \begin{cases} \max\limits_{1 \leq i \leq n, 1 \leq k \leq l_h} \left\{ h_{ij}^{\tau(k)} \right\}, & M_j \text{ is a benefit type attribute} \\ \min\limits_{1 \leq i \leq n, 1 \leq k \leq l_h} \left\{ h_{ij}^{\tau(k)} \right\}, & M_j \text{ is a cost type attribute} \end{cases} \tag{4}
$$

## 5. Decision-Making Method for Green Supply Chain Member Selection

### 5.1. Weight Determination Method

There are three main methods in attribute weight determination: subjective assignment method, objective assignment method, and combined subjective and objective assignment method [38]. In this paper, we will use the objective assignment method proposed in the literature [29], which is a hesitant fuzzy multi-attribute decision-making method considering attribute weight optimization to determine the attribute weights in green supply chain membership selection. However, Liu et al. [29] had limitations in performing weight determination and ranking of options. In this study, we will quantify each balance coefficient according to the nature of balance coefficients and rank the final decision options by introducing the three-point estimation method to further improve the scientific and rational decision making.

After comprehensive consideration of the scheme's own characteristics, attribute characteristics, and interrelationships among attributes, the following attribute weight determination method was constructed by Liu et al. [29].

$$
M \begin{cases} \max f(W) = \alpha \sum\limits_{i=1}^{n} \sum\limits_{j=1}^{m} \frac{s(h_{ij})}{s(h_{ij}) + v(h_{ij})} \omega_j + \beta \frac{1}{n} \sum\limits_{j=1}^{m} \sqrt{\sum\limits_{1 \leqslant p,q \leqslant m} d^2(h_{pj}, h_{qj})} \omega_j + \gamma \sum\limits_{j=1}^{m} \sum\limits_{t=1}^{m} (1 - r_{jt}) \omega_j \\ \text{s.t.} \sum\limits_{j}^{m} \omega_j^2 = 1, 0 \leq \omega_j \leq 1 \end{cases}
$$

where $\alpha$, $\beta$, and $\gamma$ are equilibrium coefficients, $0 \leq \alpha, \beta, \gamma \leq 1$, $\alpha + \beta + \gamma = 1$. For the values of $\alpha$, $\beta$, and $\gamma$ there exist the following cases [23]:

(1) When $\alpha = 1, \beta = \gamma = 0$, the decision maker focuses on the program's own characteristics.
(2) When $\beta = 1, \alpha = \gamma = 0$, the decision maker focuses on the attribute characteristics.
(3) When $\gamma = 1, \alpha = \beta = 0$, the decision maker focuses on the interrelationship between attributes.
(4) When $\beta \cdot \gamma \cdot \alpha \neq 0$, the decision maker integrates the program and attribute characteristics, makes full use of the decision information, and makes a more comprehensive, objective, and reasonable evaluation of the evaluation object.

For the first three cases, the values of $\alpha$, $\beta$, and $\gamma$ are determined. In addition, for the fourth case, it means that the magnitudes of $\alpha$, $\beta$, and $\gamma$ are given in advance by the decision maker according to the actual situation, so the final attribute weights will still be biased toward subjectivity.

In this paper, based on the above problem, the equilibrium coefficients $\alpha$, $\beta$, and $\gamma$ are quantified. Since the scheme $H_i = (h_{i1}, h_{i2}, h_{i3}, \ldots \ldots h_{im_i})$, $i = 1, 2, 3 \ldots n$, the larger the value in $h_{ij}$, i.e., the score value $s(h_{ij})$ is higher, the better the program is. If the variance $v(h_{ij})$ of $h_{ij}$ is smaller, i.e., the more concentrated the elements in $h_{ij}$ are, the more consistent the evaluation value provided by the decision maker is, and, thus, the smaller the impact on the decision results [29]. Therefore, we assume that in the case of (4)④, decision makers prefer to focus on the program's own characteristics, i.e., decision makers want to obtain programs that are optimal and the most consistent evaluation values provided by all decision makers, and take

$$\alpha' = \max_{1 \leq i \leq n, 1 \leq j \leq m} \frac{s(h_{ij})}{s(h_{ij}) + v(h_{ij})} \tag{5}$$

Since, for each decision scheme, if the attribute values under the attribute $M_j$ are more different or more dispersed, the greater the role played by the attribute on the decision and ranking of the scheme is, and a larger weight is assigned to the attribute at this time. Conversely, if the attribute values under the attribute $M_j$ are less different or more concentrated, the smaller the role played by the attribute $M_j$ on the decision and ranking of the scheme is, and a smaller weight is assigned to the attribute at this time [36,39]. Therefore, we assume that in case (4), the decision maker pays more attention to the attribute features and wants to assign the greatest weight to the attribute values. That is, the decision maker believes that the attribute feature is the most important and therefore should occupy the largest weight, and take

$$\beta' = \max_{1 \leq j \leq n, 1 \leq p, q \leq m} d_w(h_{pj}, h_{qj}) \tag{6}$$

where $d_w(h_{pj}, h_{qj})$ is calculated according to the hesitant Euclidean distance [40], i.e.,

$$d_w(h_{pj}, h_{qj}) = \sqrt{\frac{1}{l} \sum_{k=1}^{l} \left| h_{pj}^{\tau(k)} - h_{qj}^{\tau(k)} \right|^2}$$

Let $R_{jt}$ denote the correlation between attribute $M_j$ and the remaining attributes. The larger $R_{jt}$ is, the more similar the distribution and arrangement of attribute $M_j$ and the remaining attribute values are. when excluding attribute $M_j$, it has less influence on the scheme ranking, then attribute $M_j$ can be assigned a smaller weight. Conversely, a larger weight is assigned [39]. Therefore, we assume that in case (4), the decision maker focuses more on the interrelationship between attributes and wants to assign the highest weight to the attributes, and take

$$\gamma' = \max_{1 \leq j \leq n, 1 \leq t \leq n} (1 - r_{jt}) \tag{7}$$

where $r_{jt}$ denotes the association of attribute $M_j$ with $M_t$ and calculates $r_{jt}$ according to the following equation [41]:

$$r_{jt} = \frac{\sum_{a=1}^{m} \left( \frac{1}{l_a} \sum_{k=1}^{l_a} h_{aj}^{\tau(k)} \cdot h_{at}^{\tau(k)} \right)}{\max \left\{ \sum_{a=1}^{m} \left( \frac{1}{l_a} \sum_{k=1}^{l_a} \left( h_{aj}^{\tau(k)} \right)^2 \right), \sum_{a=1}^{m} \left( \frac{1}{l_a} \sum_{k=1}^{l_a} \left( h_{at}^{\tau(k)} \right)^2 \right) \right\}}$$

where $\alpha'$, $\beta'$, and $\gamma'$ are derived by (5)–(7), but the values of $\alpha'$, $\beta'$, and $\gamma'$ cannot be used directly in the calculation and need to be corrected according to the following three cases:

(1) When $\alpha' + \beta' + \gamma' > 1$:

$$\alpha = \alpha' - \frac{(\alpha' + \beta' + \gamma') - 1}{3}, \; \beta = \beta' - \frac{(\alpha' + \beta' + \gamma') - 1}{3}, \text{ and } \gamma = \gamma' - \frac{(\alpha' + \beta' + \gamma') - 1}{3} \tag{8}$$

If one of the values of $\alpha$, $\beta$, or $\gamma$ is less than 0, the remaining two values are corrected after excluding that value. For example, suppose $\alpha = \alpha' - \frac{(\alpha'+\beta'+\gamma')-1}{3} < 0$, then let $\theta = 1 - \alpha'$, $\alpha = \alpha'$, $\beta = \beta' - \frac{(\beta'+\gamma')-\theta}{2}$, and $\gamma = \gamma' - \frac{(\beta'+\gamma')-\theta}{2}$. If two values of $\alpha$, $\beta$, and $\gamma$ are less than 0, the remaining one is corrected by excluding those two values. For example, suppose $\alpha = \alpha' - \frac{(\alpha'+\beta'+\gamma')-1}{3} < 0$ and $\beta = \beta' - \frac{(\alpha'+\beta'+\gamma')-1}{3} < 0$, then let $\vartheta = 1 - \alpha' - \beta'$, $\alpha = \alpha'$, $\beta = \beta'$, and $\gamma = \vartheta$.

(2) When $\alpha' + \beta' + \gamma' < 1$:

$$\alpha = \alpha' + \frac{1 - (\alpha' + \beta' + \gamma')}{3}, \ \beta = \beta' + \frac{1 - (\alpha' + \beta' + \gamma')}{3}, \ \text{and} \ \gamma = \gamma' + \frac{1 - (\alpha' + \beta' + \gamma')}{3} \tag{9}$$

(3) When $\alpha' + \beta' + \gamma' = 1$:

$$\alpha = \alpha', \ \beta = \beta', \ \gamma = \gamma' \tag{10}$$

The attribute weights are [29]:

$$\omega_j^* = \frac{\alpha \sum_{i=1}^n \frac{s(h_{ij})}{s(h_{ij})+v(h_{ij})} + \beta \frac{1}{n} \sqrt{\sum_{1 \le p,q \le n} d^2(h_{pj}, h_{qj})} + \gamma \sum_{t=1}^m (1 - r_{jt})}{\sum_{j=1}^n \left( \alpha \sum_{i=1}^n \frac{s(h_{ij})}{s(h_{ij})+v(h_{ij})} + \beta \frac{1}{n} \sqrt{\sum_{1 \le p,q \le n} d^2(h_{pj}, h_{qj})} + \gamma \sum_{t=1}^m (1 - r_{jt}) \right)} \tag{11}$$

### 5.2. Decision-Making Steps

In the decision-making problem of selecting green supply chain members from the perspective of green innovation capability, let $H$, $M$, and $W$ be the decision-making scheme set for green supply chain members, the decision-making attribute set of each decision-making scheme, and the weight set of each attribute, respectively. Based on the above methods, the following decision-making steps are given:

Step 1:　The green supply chain member selection decision maker based on the green innovation capability for the set of attributes $M = \{M_1, M_2, M_3, \ldots \ldots, M_t\}$ in the solution set $H = \{H_1, H_2, H_3, \ldots \ldots, H_i\}$ is measured according to each attribute, the hesitant fuzzy decision matrix $D = \{h_{i \times t}\}$ for this green supply chain member selection is obtained, and the positive ideal solution is calculated according to Equation (4).

Step 2:　Calculate $\alpha'$, $\beta'$, and $\gamma'$ according to Equations (5)–(7), correct $\alpha'$, $\beta'$, and $\gamma'$ according to Equations (8)–(10), and determine the values of $\alpha$, $\beta$, and $\gamma$.

Step 3:　The attribute weights of the decision attributes of each decision option in the decision problem of selecting green supply chain members based on the green innovation capability perspective are determined according to Equation (11).

Step 4:　According to Formulas (2) and (3), calculate the similarity $S_O(H_i)$, $S_m(H_i)$, and $S_p(H_i)$ of each decision-making scheme and the positive ideal point in the decision-making problem of selecting green supply chain members from the perspective of green innovation capability

Step 5:　According to the three-point estimation method, the decision-making schemes in the decision-making problem of selecting green supply chain members from the perspective of green innovation ability are ranked, and the optimal green supply chain member selection scheme is determined.

Based on the above steps, we give the decision-making method for green supply chain member selection. As shown in the following Figure 1.

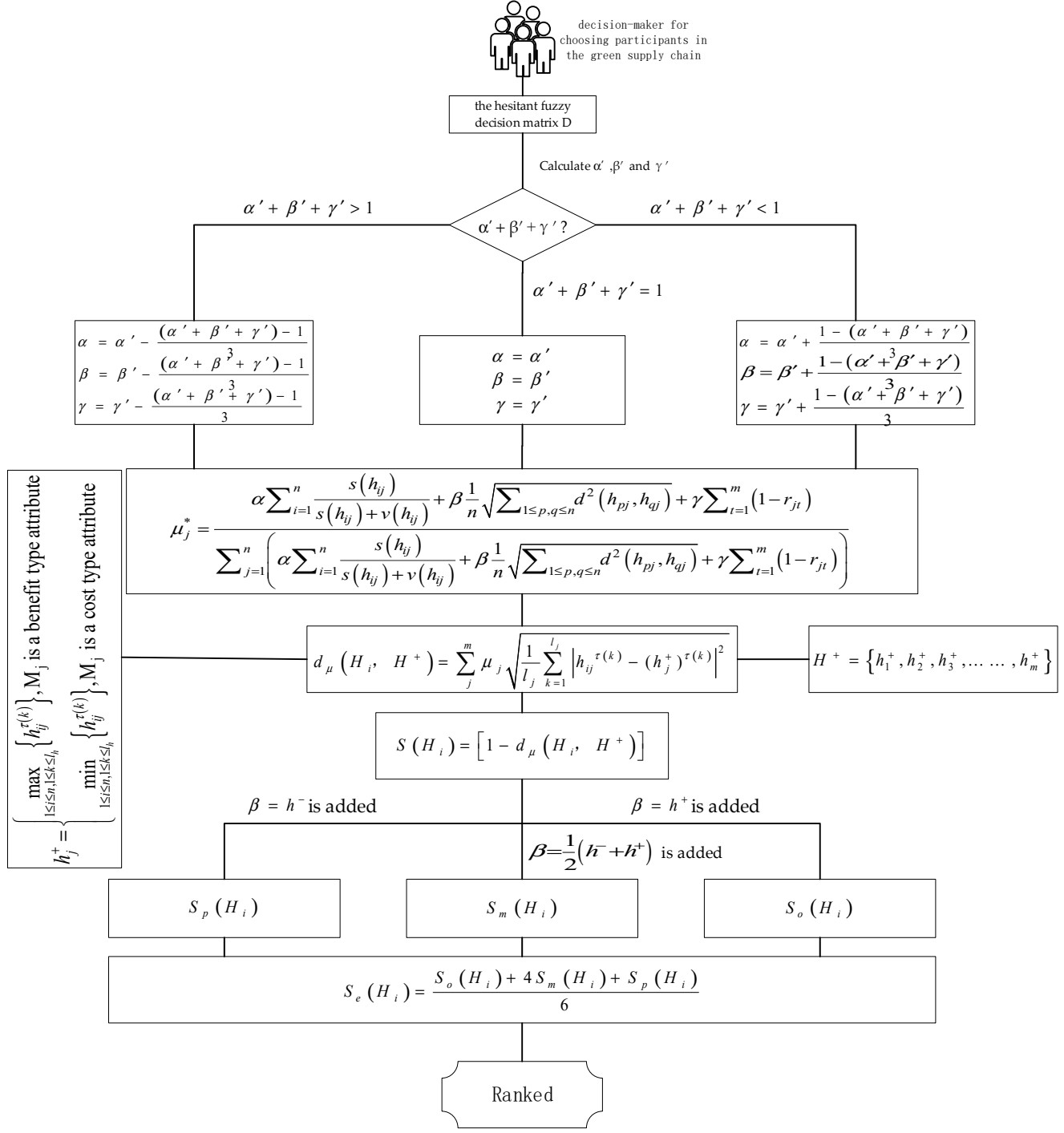

**Figure 1.** Diagram of the decision-making method for green supply chain member selection.

## 6. Numerical Example

An innovative new energy vehicle manufacturing company is a high-tech enterprise specializing in the research and development, production, sales, and service of new energy-pure electric vehicles and is a large new energy vehicle manufacturing group with "China Famous Trademark", "National High-Tech Enterprise", and "China Top 500 Private Enterprises", headquartered in Suizhou City, Hubei Province, China. The company has developed more than 10 independent innovative new energy-pure electric vehicle series products such as pure electric water sprinkler, pure electric van, pure electric refrigerator, pure electric hook-arm garbage truck, pure electric three-wheeled high-pressure cleaning truck, electric three-wheeled garbage removal truck, and pure electric closed garbage truck.

Currently, the company has decided to select the right multiple partners for its core components. Due to the specificity of the technology, the partners are expected to have some green innovation when selecting partners. The company's leadership decided to set up a special decision-making group for this decision to select green supply chain members based on green innovation capability and identified the decision-making considerations as green innovation input, subject synergy, green innovation output capability, system innovation capability, and green innovation sustainability. In addition, considering the company's large business in core components, the company decided to select two companies to cooperate accordingly. After evaluation and selection, the decision-making team finally selected five cooperative companies (solutions) to choose from, mainly considering five factors, green innovation input $M_1$, main body synergy $M_2$, green innovation output capacity $M_3$, system innovation capability $M_4$, and sustainability of green innovation $M_5$, where $M_1$ are cost-based attributes and the rest are benefit-based attributes. After that, the decision panel analyzed the program $H_i(i = 1, 2, 3, \ldots \ldots m)$ by each attribute $M_j = \{j = 1, 2, 3, \ldots \ldots, n\}$ and obtained the hesitant fuzzy decision matrix $D = (h_{ij})_{5 \times 5}$ (see Table 1), where $h_{11} = \{0.2, 0.4, 0.7, 0.9\}$ indicates that there are four different views of the decision group in terms of green innovation inputs, i.e., 0.2, 0.4, 0.7, and 0.9 for the extent to which option $H_1$ satisfies attribute $M_1$, which means that the members in the decision group have different views. When a decision maker is unfamiliar with an attribute, that decision maker may not make a decision on that attribute, or several decision makers appear to have the same decision value when making a decision on the same attribute, these situations result in inconsistent elements in the fuzzy elements. For example, the number of elements in the hesitation fuzzy elements of scenario $H_1$ under attributes $M_3$ and $M_4$ is inconsistent.

**Table 1.** Green supplier decision matrix $D = \left(h_{ij}\right)_{5 \times 5}$ for new energy vehicle manufacturers.

| Program | $M_1$ | $M_2$ | $M_3$ | $M_4$ | $M_5$ |
|---|---|---|---|---|---|
| $H_1$ | {0.2, 0.4, 0.7, 0.9} | {0.1, 0.2, 0.5, 0.7} | {0.2, 0.3, 0.5, 0.7, 0.8} | {0.1, 0.4, 0.6} | {0.3, 0.5, 0.7, 0.8} |
| $H_2$ | {0.4, 0.6, 0.7, 0.8} | {0.1, 0.2, 0, 4, 0.6} | {0.3, 0.4, 0.7, 0.8, 0.9} | {0.1, 0.2, 0.4} | {0.4, 0.5, 0.7, 0.9} |
| $H_3$ | {0.2, 0.3, 0.6, 0.7} | {0.3, 0.4, 0.5, 0.7} | {0.2, 0.4, 0.6, 0.7, 0.8} | {0.3, 0.4, 0.8} | {0.2, 0.6, 0.7, 0.8} |
| $H_4$ | {0.2, 0.3, 0.5, 0.8} | {0.2, 0.3, 0.5, 0.7} | {0.4, 0.6, 0.7, 0.8, 0.9} | {0.1, 0.2, 0.7} | {0.3, 0.5, 0.8, 0.9} |
| $H_5$ | {0.2, 0.3, 0.4, 0.7} | {0.3, 0.4, 0.6, 0.7} | {0.1, 0.5, 0.6, 0.8, 0.9} | {0.2, 0.5, 0.6} | {0.2, 0.4, 0.6, 0.9} |

Calculate the positive ideal point of this green supply chain member selection option based on Equation (4).

$$H^+(\{0.2\}, \{0.7\}, \{0.9\}, \{0.8\}, \{0.9\})$$

The equilibrium coefficients $\alpha'$, $\beta'$, and $\gamma'$ are calculated according to Equations (5)–(7) and corrected according to Equations (8)–(10) until the final equilibrium coefficients $\alpha$, $\beta$, and $\gamma$.

$\alpha' = 0.658$, $\beta\prime = 0.282$, and $\gamma' = 0.172$. After correction, $\alpha = 0.621$, $\beta = 0.245$, and $\gamma = 0.134$.

The weights of each attribute are determined according to Equation (11).

$$W = (0.201, 0.193, 0.175, 0.214, 0.207)^T$$

Calculate the similarity of each decision scheme to the positive ideal point according to Equations (2) and (3): $S_O(H_i)$, $S_m(H_i)$, and $S_p(H_i)$. The results are shown in Table 2.

**Table 2.** Similarity of $S_O(H_i)$, $S_m(H_i)$, and $S_p(H_i)$.

| Program | $S_O(H_i)$ | $S_m(H_i)$ | $S_p(H_i)$ |
|---------|-----------|-----------|-----------|
| $H_1$ | 0.657 | 0.732 | 0.788 |
| $H_2$ | 0.692 | 0.736 | 0.774 |
| $H_3$ | 0.717 | 0.78 | 0.828 |
| $H_4$ | 0.660 | 0.74 | 0.796 |
| $H_5$ | 0.670 | 0.734 | 0.785 |

Calculate the expected similarity of each solution according to the three-point estimation method and rank the solutions according to the expected similarity. The results are shown in Table 3.

**Table 3.** Decision-making results and program ranking.

| Program | $S_e(H_i)$ | Three-Point Estimation Method | Program Ranking (Excellent → Poor) |
|---------|-----------|-------------------------------|-----------------------------------|
| $H_1$ | 0.729 | | $H_3$ |
| $H_2$ | 0.735 | | $H_4$ |
| $H_3$ | 0.777 | $S_e(H_i) = \frac{S_o(H_i)+4S_m(H_i)+S_p(H_i)}{6}$ | $H_2$ |
| $H_4$ | 0.736 | | $H_5$ |
| $H_5$ | 0.732 | | $H_1$ |

According to the above calculation results, the recommended ranking of the decision options given by the decision panel after considering all the attributes is: $H_3 \succ H_4 \succ H_2 \succ H_5 \succ H_1$, so the two options $H_3$ and $H_4$ are recommended to the company; that is, the two options $H_3$ and $H_4$ are selected to correspond to the company as the subsequent green supply chain partner.

## 7. Comparative Analysis

For a hesitant fuzzy multiple attribute decision-making problem with unknown attribute weights, Liu et al. [29] proposed a hesitant fuzzy multiple-attribute decision-making method considering attribute weight optimization, Chen et al. [30] proposed an improved TODIM hesitant fuzzy multiple-attribute decision-making method considering weight optimization, Zhang et al. [31] proposed an objective attribute weights determining based on Shannon information entropy in hesitant fuzzy multiple-attribute decision making, this part will be analyzed by comparing the method proposed in this paper with the methods proposed by the above scholars.

Since previous subjective assignment methods relied heavily on the perceived assignment of weights to attributes, the literature [29] considered the dispersion of decision information (i.e., variance) and correlations among attributes along with the scores of the solutions under each attribute, thus creating an optimization model to determine the weights of the attributes. However, this method favors subjective considerations in determining the equilibrium coefficients, and, in the method proposed by Liu et al. [29], the value of the equilibrium coefficients directly affects the values of the attribute weights, which ultimately has a direct impact on the decision results. In addition, it will directly lead to the bias of the decision result due to the inaccurate judgment of the decision maker in determining the equilibrium coefficient, and it can also result in the loss of the benefits of decisions in other risk scenarios due to a single consideration of risk appetite (see Table 4 for the comparison of the results). However, the improved method proposed in this paper can well avoid the shortcomings of the method in the literature [19]. By quantifying and correcting the equilibrium coefficients, this paper makes the equilibrium coefficients more objective and avoids affecting the decision results due to subjective negligence. Meanwhile, by introducing the three-point estimation method into the ranking of decision results, this paper integrates the influence of different s on the decision results and makes the decision results uniform, which is conducive to decision makers' selection of decision options.

**Table 4.** Comparison with the method in the literature [29].

| Decision-Making Methods | Balance Factor | Attribute Weights | Program Sorting |
|---|---|---|---|
| Program Sorting | $\alpha = 1$ <br> $\beta = 0$ <br> $\gamma = 0$ | $(0.200, 0.194, 0.176, 0.210, 0.210)^T$ | $H_3 \succ H_2 \succ H_5 \succ H_4 \succ H_1 (p)$ <br> $H_3 \succ H_4 \succ H_5 \succ H_1 \succ H_2 (m)$ <br> $H_3 \succ H_4 \succ H_1 \succ H_5 \succ H_2 (o)$ |
| | $\alpha = 0$ <br> $\beta = 1$ <br> $\gamma = 0$ | $(0.256, 0.141, 0.163, 0.335, 0.105)^T$ | $H_5 \succ H_2 \succ H_3 \succ H_1 \succ H_4 (p)$ <br> $H_5 \succ H_3 \succ H_2 \succ H_1 \succ H_4 (m)$ <br> $H_5 \succ H_3 \succ H_2 \succ H_1 \succ H_4 (o)$ |
| | $\alpha = 0$ <br> $\beta = 0$ <br> $\gamma = 1$ | $(0.122, 0.261, 0.165, 0.287, 0.168)^T$ <br> $(0.109, 0.250, 0.176, 0.265, 0.173)^T$ <br> $(0.106, 0.192, 0.226, 0.245, 0.212)^T$ | $H_1 \succ H_3 \succ H_5 \succ H_2 \succ H_4 (p)$ <br> $H_1 \succ H_3 \succ H_5 \succ H_4 \succ H_2 (m)$ <br> $H_1 \succ H_3 \succ H_5 \succ H_4 \succ H_2 (o)$ |
| | $\alpha = 1/3$ <br> $\beta = 1/3$ <br> $\gamma = 1/3$ | $(0.200, 0.194, 0.173, 0.222, 0.201)^T$ <br> $(0.200, 0.193, 0.174, 0.220, 0.202)^T$ <br> $(0.200, 0.191, 0.176, 0.219, 0.203)^T$ | $H_3 \succ H_2 \succ H_5 \succ H_1 \succ H_4 (p)$ <br> $H_3 \succ H_5 \succ H_2 \succ H_1 \succ H_4 (m)$ <br> $H_3 \succ H_5 \succ H_1 \succ H_4 \succ H_2 (o)$ |
| Program Sequencing for this Article | $\alpha = 0.621$ <br> $\beta = 0.245$ <br> $\gamma = 0.134$ | $(0.201, 0.193, 0.175, 0.214, 0.207)^T$ | $H_3 \succ H_4 \succ H_2 \succ H_5 \succ H_1$ |

Note. In the table, *o* denotes the risk-appetite scenario, *m* denotes the risk-neutral scenario, and *p* denotes the risk-averse scenario.

In the literature [30], a multi-objective optimization model was constructed from two levels of solution and attribute values with hesitant fuzzy number mean, variance, and non-explicit entropy to determine the relative weights. At the same time, considering the different psychological behavioral characteristics of decision makers, the perceived value function matrix of each attribute was obtained based on the value function of the relative "gain" or "loss" values under the two-comparison of solutions. Through the simple weighting principle, the perceived value function matrix and the relative weights are assembled to give a new expression of the integrated perceived value function and the global dominance, the solutions are ranked based on the expression of the global dominance, and the ideal solution is finally selected. Chen et al. [30] improves the original TODIM method to a certain extent and makes full use of decision information. However, when calculating the perceived value function, the coefficients of the decision maker's sensitivity to "gain" or "loss" and the coefficients of psychological characteristics directly affect the final decision result. In other words, different coefficients can lead to different decision results, which requires very high coefficient values. If the coefficients cannot be quantified by an objective method, the decision maker may not obtain the optimal decision result or may even deviate from the decision maker's original intention. Therefore, this study overcomes the shortcomings of the previous literature in the coefficient processing through objective methods and the coefficient quantification by decision information, and corrects the processed results very well. Through comparison, it can be found that there will be some differences in the ranking between the research method of this paper and the research method in the literature [30], but $H_3$ is always ranked first (see Tables 5 and 6).

**Table 5.** Combined perceived value function values $H_i$ for scenario relative $H_v$.

| | $H_1$ | $H_2$ | $H_3$ | $H_4$ | $H_5$ |
|---|---|---|---|---|---|
| $H_1$ | 0 | −0.203 | 0.146 | −0.154 | 0.012 |
| $H_2$ | −0.428 | 0 | −0.198 | −0.037 | −0.410 |
| $H_3$ | −0.805 | −0.756 | 0 | −0.348 | −0.314 |
| $H_4$ | −0.443 | −0.699 | −0.268 | 0 | −0.315 |
| $H_5$ | −0.703 | 0.516 | −0.125 | −0.244 | 0 |

**Table 6.** Global dominance degree and comparison with the method in the literature [30].

| | $H_1$ | $H_2$ | $H_3$ | $H_4$ | $H_5$ |
|---|---|---|---|---|---|
| Global Dominance Degree | 0 | 0.106 | 1 | 0.826 | 0.699 |
| Program Sorting | | | $H_3 \succ H_4 \succ H_5 \succ H_2 \succ H_1$ | | |
| Program Sequencing for this Article | | | $H_3 \succ H_4 \succ H_2 \succ H_5 \succ H_1$ | | |

In the literature [31], a new objective attribute weighting method based on Shannon information entropy was proposed to express the relative strength of attribute importance and determine objective attribute weights. Unlike previous decision-making methods, Zhang et al. [31] uses Shannon information entropy to express the relative strength of attribute importance and determine objective attribute weights, rather than using hesitant fuzzy entropy to determine attribute weights based on the credibility of the input data. Therefore, the method proposed in the literature [31] can well avoid the incorrect decision results caused by the inaccuracy or low confidence of the hesitant input data. However, in the literature [31], two ranking results are given when the final solution ranking is performed, which makes the two decision results inconsistent in the solution ranking in some extreme cases, causing the decision maker to be unable to make a decision. Thus, this study provides results for three scenarios of risk aversion, risk preference, and risk neutrality for the similarity of each solution in the solution ranking, calculates the expected similarity based on the three-point estimation method, and finally provides a unique solution ranking for the decision maker to choose based on the expected similarity. The comparison shows that the decision-making method in this paper is different from that of Zhang et al. [31], but the decision results are the same when the best solution is selected (See Table 7).

**Table 7.** The correlation coefficients and ranking results of all the alternatives.

| | $C(H_1,H_0)$ | $C(H_2,H_0)$ | $C(H_3,H_0)$ | $C(H_4,H_0)$ | $C(H_5,H_0)$ |
|---|---|---|---|---|---|
| $C\vert_{\lambda=1}$ | 0.759 | 0.515 | 0.903 | 0.731 | 0.838 |
| $C\vert_{\lambda=2}$ | 0.939 | 0.744 | 0.989 | 0.916 | 0.973 |
| Program Sorting | | | $H_3 \succ H_5 \succ H_1 \succ H_4 \succ H_2$ | | |
| Program Sequencing for this Article | | | $H_3 \succ H_4 \succ H_2 \succ H_5 \succ H_1$ | | |

By comparing the method of this study with the existing solutions, it can be found that the ranking of the solutions will be different depending on the hesitation method system and the calculation of the attribute weights, but the first ranked solution is the same. In the green supply chain, for some large enterprises, when selecting partners based on green innovation capability, it often appears that different green supply chain members are particularly innovative in different aspects and relatively weak in other aspects. Therefore, in the selection of green supply chain members considering green innovation capability, it is necessary to consider the benefits brought by different risk preferences, and it may be necessary to select multiple members to participate in the green supply chain. Based on these considerations, the improvement method proposed in this paper can well meet the selection of green supply chain members considering green innovation capability.

## 8. Conclusions

In this paper, the selection of green supply chain members is studied from the perspective of green innovation capability. By summarizing the evaluation indices of innovation capability in previous literature, this paper proposes green innovation input, the synergy of subjects in the green supply chain, green innovation output capability, the institutional innovation capability of enterprises in the green supply chain, and green innovation sustainability to evaluate the green innovation capability of green supply chain members. Since different index attributes need to be used to select different options, there is often some hesitation and fuzziness in the selection process. Moreover, in decision making,

this paper is written to evaluate multiple alternative members by multiple indicator attributes and then rank the alternative members for decision making. Therefore, to be able to make full use of the available decision information, this paper adopts the hesitant fuzzy multi-attribute decision-making method that considers attribute weight optimization as proposed in the literature [29] for green supply chain membership selection considering green innovation capability based on hesitant fuzzy set theory. However, there are two methodological shortcomings in the literature [29]: first, the balance coefficients used to calculate attribute weights are artificially and subjectively chosen; second, the ranking of scenarios only takes into account one type of risk preference while ignoring the situation in which decision makers are unable to determine their risk type. Therefore, on the one hand, this paper carries out an objective decision on the balance coefficient by quantifying the balance coefficient to ensure that the final attribute weights are determined objectively. On the other hand, this paper solves the decision-making problem that decision makers cannot determine their risk preference types by calculating the solution evaluation values under three scenarios, risk-pessimistic, risk-neutral, and risk-optimistic, introducing the three-point estimation method to synthesize the solution evaluation ideal values under the three risk preferences, and then ranking the solutions according to the solution ideal values. This paper also applies this decision-making method to a case of selecting spare parts suppliers for an innovative new energy vehicle manufacturer in Suizhou City, Hubei Province, China. The methods proposed in the literature [29–31] are also compared with the method in this paper, and the analysis results show that the decision results of the three methods are consistent, which also indicates that the decision method we give is scientific and reasonable.

The research value of this paper can be summarized as follows. First, this paper aims at green supply chain membership selection under the vision of "green innovation capability", improves the deficiencies of the literature [29], provides some theoretical reference for the green supply chain membership selection problem considering "green innovation capability", and provides some practical value for solving similar hesitant fuzzy multi-attribute decision problems in other fields. Secondly, this paper identifies the index attributes to evaluate the scheme from the perspective of green innovation capability and uses a hesitant fuzzy set as an information expression tool to aggregate the decision information of each attribute, which provides a research idea for the selection of green supply chain members. Finally, this paper also provides a reference for hesitant fuzzy sets in attribute weight determination and makes a certain contribution to the application of hesitant fuzzy sets.

Through example analysis, the method of this paper has high practicality in the hesitant fuzzy decision environment of green suppliers, with wide application and closer to practical decision making. In future research, we will consider extending the study of this paper to probabilistic hesitation fuzzy numbers. Although the hesitation fuzzy number can aggregate all the decision values of all decision makers for the same solution, it is only able to aggregate one value when there are decision makers with the same decision value, which ignores the importance of the decision value. However, the probabilistic hesitation fuzzy number can handle such a situation in a probabilistic form. In addition, in future research, we will consider applying the method of this paper to some other decision problems, e.g., project site selection, large equipment purchase, etc.

**Author Contributions:** Conceptualization, L.L. and J.S.; methodology, J.S. and B.X.; software, B.X.; formal analysis, D.W. and B.X.; investigation, F.Z.; resources, L.L.; writing—original draft preparation, D.W. and B.X.; funding acquisition, J.S. All authors have read and agreed to the published version of the manuscript.

**Funding:** This work was supported by the Natural Science Foundation of Chongqing Science & Technology Commission (Grant No. CSTB2022NSCQ-MSX0478) and the Technology Research Program of Chongqing Municipal Education Commission (Grant No. KJQN202200808).

**Data Availability Statement:** The data used to support the findings of this study are included within the article.

**Conflicts of Interest:** The authors declare no conflict of interest.

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
