# Peer review of "A Green Supply Chain Member Selection Method Considering Green Innovation Capability in a Hesitant Fuzzy Environment"

_axioms, doi:10.3390/axioms12020188_

Round 1

Reviewer 1 Report

The paper presents an updated method for the multi-criteria evaluation of suppliers based on the concepts of hesitant fuzzy sets and objective weights. While the authors focus their attention on the selection of innovative suppliers as potential members of a green supply chain, the paper is mainly methodological and the case study gives only limited information about the evaluation performed with regard to such subject.
The presentation of the methodology is confusing in some places, and it surely needs a base of knowledge on hesitant fuzzy sets to be understood.
The paper is potentially interesting for researchers in multicriteria decision methods, but it could be hard to follow by practitioners of the industrial sector. In fact, the less expert may not have all the requirements to appreciate the methodological contribution.
The paper structure is non always well defined and some sections could be organised more effectively. The English form is mostly correct with some random mistakes, but the fluency of some paragraphs can be improved. The specific comments are included in the attached pdf: addressing the questions reported there could help to improve the paper.

Author Response

Response to Reviewer 1 Comments:

Overall comments: The paper presents an updated method for the multi-criteria evaluation of suppliers based on the concepts of hesitant fuzzy sets and objective weights. While the authors focus their attention on the selection of innovative suppliers as potential members of a green supply chain, the paper is mainly methodological and the case study gives only limited information about the evaluation performed with regard to such subject.

The presentation of the methodology is confusing in some places, and it surely needs a base of knowledge on hesitant fuzzy sets to be understood.

The paper is potentially interesting for researchers in multicriteria decision methods, but it could be hard to follow by practitioners of the industrial sector. In fact, the less expert may not have all the requirements to appreciate the methodological contribution.

The paper structure is non always well defined and some sections could be organised more effectively. The English form is mostly correct with some random mistakes, but the fluency of some paragraphs can be improved. The specific comments are included in the attached pdf: addressing the questions reported there could help to improve the paper.

Response : First of all, thank you for your valuable and useful comments on our manuscript. Our research is to propose an innovative hesitant fuzzy multi-attribute decision making method for green supply on selection under green innovation capability. We have spent a lot of work on the innovation of the method, so the information on the green supply chain seems relatively insufficient. However, this does not affect the application of our method to the selection of green supply chain members under the green innovation capability perspective. As you say, our study may be interesting to researchers of multi-criteria decision making methods, but may be difficult to understand for practitioners in the industrial sector. This is because we use an information expression representation tool for hesitant fuzzy sets, which requires some expertise to be well understood. Nevertheless, the study of hesitant fuzzy sets has received a lot of scholarly attention and has been widely used in many fields. This is because hesitant fuzzy sets are very effective in expressing information in a fuzzy environment. Of course, there are some linguistic and structural deficiencies in our paper, which make our manuscript somewhat confusing.

Moreover, we have carefully revised our manuscript by summarizing and reflecting on the valuable comments you provided in the PDF, and this has led to a significant improvement in our manuscript. The specific revisions are as follows:

Point 1: The manuscript has problems with "inappropriate use of words, unclear and incomplete sentences, unclear paragraphs, etc." For example, “This phrase should be revised as it is not very clear.”, “‘goods or services’ is better here”, “This part is not very clear.” and so on.

Response 1: Thank you very much for providing such detailed comments on our manuscript, which have been very valuable to us in improving our manuscript. We have carefully followed your comments to make revisions to the corresponding parts in the text, and since there are many contents, we will not show them all here, you can check them in the revised manuscript.

Point 2: This paragraph is a bit confusing. You should revise it. In particular, it is not clear what do you mean with ‘selection of members’: is there a main player in the chain? Who is responsible for the selection of the suppliers involved in the supply chain? What control mechanisms can be put into place to make the SC ‘green’?

Response 2: This comment is very valuable. The “select members” here mainly refers to the mutual selection among the companies in the green supply chain, not limited to the selection of suppliers, but also the selection of retailers, manufacturers and so on. And we mainly use the green-related indicators proposed in this paper to decide whether it is "green" or not. In the second paragraph of the introduction, we have added some content to explain these confusions. The details are as follows:

“And in order to accomplish coordination and optimization of economic and social benefits, each firm in the supply chain and its internal departments must work closely together.” “Whether it is suppliers, manufacturers, distributors, retailers, or end-users, there is a mutual choice between them. For the whole supply chain, whether it is a matter of suppliers choosing manufacturers or manufacturers choosing suppliers, or manufacturers choosing distributors or distributors choosing retailers, it is a matter of member selection in the supply chain.”

Point 3: Are end customers considered part of the SC? Is a selection stage applied to them too?

Response 3: We are grateful to your suggestions that improved our manuscript greatly. The fact that the end customer was not part of the selection for our study was not clearly expressed and we have revised it in the manuscript. The specific revisions are as follows:

“We select green innovation capability as the criterion for evaluating members., and the selection of members is not limited to suppliers and manufacturers, but involves companies in the whole supply chain.”

Point 4: Some different aspects of multi-criteria are mixed up here: performance assessment, by means of HFN, and weighing of criteria. You have not introduced the MCDM method you will use, so the role of weights remains unclear up to this point.

Response 4: We are grateful to your suggestions that improved our manuscript greatly. This problem was caused by our lack of clarity in the manuscript, and we have revised this paragraph in the introduction. The details can be found in the fourth paragraph of the introduction.

Point 5: What do you want to highlight here? A distinction between ‘members’ and ‘suppliers’?

Response 5: This comment is very valuable. What we want to emphasize here is the difference between members and suppliers, because the members we study are not just limited to suppliers but the selection relationships between companies in the whole supply chain except for the end users.

Point 6: Compared to what? Why, among many papers about MCDM selection methods of suppliers, you selected this one?

Response 6: Thanks for the valuable comment and suggestion.This literature is not only a literature on supplier MCDM selection methods, but also the literature considers decision mechanisms in a fuzzy environment, which is very relevant to our study, so we have chosen it for reference. However, the content bias caused by our unclear exposition of the literature has been revised in our manuscript. The details are as follows:

“Abdullah et al. [15] found that when using this method for green supply chain supplier selection, they can find the best green suppliers for green supply chain management better than traditional methods.”

Point 7: Have you found any source that supports this claim?

Response 7: This comment is very valuable. Such a statement is too one-sided, so we have revised it in the manuscript and found relevant literature to support our view. The details are as follows:

“However, green innovation capabilities can be well integrated into these three aspects [17].”

Point 8: Why do you think that this case study is relevant?

Response 8: Thanks for the valuable comment and suggestion. We have re-read this literature carefully and found that the study is marginally relevant to our study, but the relevance is not strong, so we have removed the literature and the relevant content from the manuscript.

Point 9: You seem to stress that the evaluation of ‘partners’ performed by different actors (or experts) makes the decision problem ‘indecisive’ (I think you mean the result of the application of a method, more than the ‘problem’) and calls for a fuzzy approach: can you better define these cause-effect

Response 9: This comment is very valuable. We have revised this section in the manuscript. The details are as follows:

“However, the above literature does not take into account the fact that the same decision indicator will be scored by different decision makers in the partner selection process. Thus, when different decision makers score each indicator, there will be a situation where one indicator is associated with multiple scoring values, i.e., different decision makers will give different decision values under one attribute.” “The aforementioned literature tends to prefer these decisions as a simple one when making decisions; however, in reality, the selection of members in a supply chain is a complex choice process that requires the evaluation of possible courses of action or options by selecting a preferred option or ranking the options from best to worst, which is a process of multi-attribute decision analysis. In daily practice, the use of MADA is essential to signal the best rational choice to the decision maker so that he can allocate limited resources among competing and alternative interests [27].”

Point 10: The concept of ‘optimised weights’ here is not so clear, and it seems to refer to different aspects concerning weights.

Response 10: Thanks for the valuable comment and suggestion. We have added a detailed description of "Optimization weights" to our manuscript. The details are as follows:

“The above-mentioned studies use quantitative methods to optimize the attribute weights, and then combine those weights with decision-making methods to apply the optimized weights to decision-making problems. As a result, they avoid the shortcomings of subjective weight assignment, which is heavily dependent on the decision-maker due to its subjectivity.”

Point 11: As a reader, I cannot understand from this paragraph the structure of your proposal. The stages of your approach are not presented in a way that is comprehensible using the information you gave above. For instance, why determining the equilibrium coefficient is the first step of the procedure? Why do you select the ranking method after determining the equilibrium? Etc.

Response 11: This comment is very valuable. This confusion is caused by our lack of clarity in the manuscript, where we have provided some clarification of the approach proposed in this paper. The details are as follows:

“In summary, by processing the equilibrium coefficient method and solution ranking method in the decision making method based on the literature [29], this paper proposes an improved hesitant fuzzy multi-attribute decision making method for green supply chain member selection taking attribute weight optimization under the perspective of green innovation capability.” “The specific steps of the method can be found in section 5.2.”

Point 12: How can you define ‘a satisfactory decision result’?

Response 12: Thanks for the valuable comment and suggestion. Regarding "a satisfactory result", we refer to the result obtained by combining the decision information of all decision makers using hesitant fuzzy sets and then ranking each option by our proposed method. We may not have explained this clearly in the manuscript. Therefore, we have revised the statement in the manuscript. The exact formulation is as follows:

“In a realistic green supply chain member selection, decision makers always face various uncertainties, which make them hesitate to evaluate the selection of decision objects, thus making it difficult to obtain a decision result that combines information from all decision makers.”

Point 13: This is not clear: l_h is the number f elements in the ‘element’ h? Therefore l_h is an integer (or can it be a real) number? What is the relation between h and H?

Response 13: This comment is very valuable. Usually  is used, but in our manuscript , so there is no correspondence between H and h. But in the later calculation we use H to denote a scheme set and h to denote hesitant fuzzy elements. About "l_h is the number of f elements in the "element" h? Therefore l_h is an integer (or it can be a real number)?” ,the current literature related to hesitant fuzzy numbers is represented as in our manuscript, and we are referring to such literature to represent it.

Point 14: What do you mean with ‘superior’? Perhaps ‘preferable’? But this is so only when the objective is maximise the value of s(.)

Response 14: This comment is very valuable. This problem is the result of our wording errors, which we have indicated in the manuscript with symbols. The details are as follows:

“if , then ; if , then ; if , then .”

Point 15: You should define gamma…

Response 15: This comment is very valuable. We have defined gamma. The details are as follows:

“where g is the element in h”

Point 16: As you possibly noticed, this equation represents the ‘graded mean integration’ for a triangular fuzzy number (see Chen, Shan Huo, and Chih Hsun Hsieh. 2000. “Representation, Ranking, Distance, and Similarity of L-R Type Fuzzy Number and Application.” Australian Journal of Intelligent Processing Systems 6 (4): 217–29.)

Response 16: Thank you very much for recommending this paper. We have read this paper carefully and it has been very helpful to us. But our manuscript is mainly an application of the method in its most original sense.

Point 17: Do you mean ‘HF takes into consideration three risk attitudes’? But, are the values only related to risk, not for instance ‘confidence’ in the estimation?

Response 17: This comment is very valuable. HF does not take into account the three risk attitudes; it is simply a tool for expressing information. We add elements to the HF by considering the three risk attitudes, and then synthesize the risk by the three-point estimation method. So there is no such situation as you mentioned.

Point 18: It is necessary that you clearly define the meaning of the used symbols, otherwise only who already knows the methodology can understand your paper. Specifically, the differences between H and h can only be guessed…

Response 18: Thanks for the valuable comment and suggestion. We have added to the manuscript a description of the relationship between H and h. The details are as follows:

“We designate the scenario by H, and h denotes a set of evaluations of the situation H made by the decision-maker, i.e. hesitant fuzzy elements.”

Point 19: It seems that you use (2) to rank the alternatives: is it so? Please, be more clear. You haven’t reported in the previous sections that you are using a method based on the distance from an ideal solution (as it seems here). Further, omega seems to be the weight of a criterion, but this is not introduced in the text and in section 4 you use mu to indicate the weight…

Response 19: Thanks for the valuable comment and suggestion. We have modified the formula in the manuscript. The details are as follows:

“ ”

Point 20: Where is this formula? Why do you assess the weights after the application of HF sets?

Response 20: This comment is very valuable. We have revised the original presentation. The details are as follows:

“The similarity of each scenario in the risk-optimistic, risk-neutral, and risk-pessimistic scenarios is then calculated based on the attribute weights and the decision matrix, using the formula for obtaining the attribute weights following parameter optimization provided in Section 5.1 of this paper. Finally, the alternatives are ranked according to equations (1), (2), and (3),”

Point 21: This is debatable: you can have values with low variance but without knowing the true value.

Response 21: This comment is very valuable. By deliberating in the next time, we have revised in the manuscript. The details are as follows:

“thus the smaller the impact on the decision results”

Point 22: Please, explain this point better.

Response 22: We are grateful to your suggestions that improved our manuscript greatly. We have explained the corresponding contents in the manuscript. The details are as follows:

“Therefore, we assume that in case â‘£, the decision maker pays more attention to the attribute features and wants to assign the greatest weight to the attribute values. That is, the decision maker believes that the attribute feature is the most important and therefore should occupy the largest weight,”

Point 23: Do you mean ‘need to be normalized using equations (8)’?

Response 23: We are grateful to your suggestions that improved our manuscript greatly. This was not clearly expressed in the manuscript and we have revised the manuscript. The details are as follows:

“but the values of , ,  cannot be used directly in the calculation and need to be corrected according to the following three cases:”

Point 24: Can you just explain why some attributes have more elements than others? Can you just show how you identify the ideal (H+) because of the different number of elements?

Response 24: This comment is very valuable. The ideal solution (H+) is determined by equation (4), which is described in the problem description section of the manuscript. Because when evaluating an indicator attribute, the same value may appear or some decisions in which the attribute is unfamiliar cannot be evaluated, there will be some attributes with more elements than others. We have added this explanation to the arithmetic example. The details are as follows:

“When a decision maker is unfamiliar with an attribute, that decision maker may not make a decision on that attribute; or several decision makers appear to have the same decision value when making a decision on the same attribute, and these situations result in inconsistent elements in the fuzzy elements. For example, the number of elements in the hesitation fuzzy elements of scenario H1 under attributes M3 and M4 is inconsistent.”

Point 25: The meaning of the values is not explained in terms of green innovation capability, so the case study doesn’t provide a clear connection with the context of your proposal, and this might reduce the interest of researchers in the field of green supply chain planning.

Response 25: This comment is very valuable. We have revised it in the manuscript. The details are as follows:

Table 1. Green supplier decision matrix  for new energy vehicle manufacturers.”

Point 26: If the decision has consequences on a decision maker’s activity or wealth it is evident that the selected decision depends on her/his risk attitude. An ‘objective’ risk attitude doesn’t exist. When you propose to maximise the aspects that are expressed by alpha, beta and gamma, you decide for a specific risk profile; it can be good for many decision makers but not for everyone.

Response 26: Thank you very much for your valuable comments, we have thought about this insight of yours, but our approach is to want to synthesize the risk when sorting the final solution. Because facing high return means high risk, and low risk means low return, we want to achieve a desired return through such a treatment of risk synthesis. At the same time, some people cannot determine their risk type when they do not know the market very well, thus making it impossible to make rational decisions, so that the methods in this paper can help in decision making.

Point 27: Perhaps you could revise the Conclusion after updating the contents of your paper.

Response 27: We are grateful to your suggestions that improved our manuscript greatly. We have extensively revised the conclusion section to provide more details of the article and to detail the strengths of the paper's methodology and the value of the paper's research. The specific revisions can be found in the conclusion section of the manuscript.

Reviewer 2 Report

Dear Authors

With the usual compliments. I congratulate you on your submitted research proposal. The topic is relevant, the paper is well structured, and the results are relevant. However, I suggest that some points be improved, as described below:

1. I suggest that the authors insert future research gaps in the abstract;

2. In section 2. the authors addressed the aspects related to multicriteria methods. I suggest that the authors broaden the discussion about multicriteria methods, indicating the main methods and their applications related to the paper's main theme. Recently a review of multicriteria methods was published. I suggest its reading and inclusion because it can help in this discussion: https://doi.org/10.3390/electronics11111720;

3. Revise the titles of the tables and format them according to the MDPI author's guide;

4. The conclusion should be more detailed and better address the methods' advantages.

Good review.

Best regards,

Reviewer

Author Response

Response to Reviewer 2 Comments:

Point 1: I suggest that the authors insert future research gaps in the abstract.

Response 1: This comment is very valuable. We have inserted future research gaps in the abstract, as follows:

“In future research, the method proposed in this paper can be considered to combine with intuitionistic fuzzy sets, probabilistic hesitant fuzzy sets and some other fuzzy sets for method extensions to solve multi-attribute decision making problems.”

Point 2: In section 2, the authors addressed the aspects related to multi-criteria methods. I suggest that the authors broaden the discussion about multi-criteria methods, indicating the main methods and their applications related to the paper's main theme. Recently a review of multi-criteria methods was published. I suggest its reading and inclusion because it can help in this discussion: https://doi.org/10.3390/electronics11111720;

Response 2: We are grateful to your suggestions that improved our manuscript greatly. We have added a discussion of the multi-criteria approach in the fourth paragraph of the literature review. The details are as follows:

“The aforementioned literature tends to prefer these decisions as a simple one when making decisions; however, in reality, the selection of members in a supply chain is a complex choice process that requires the evaluation of possible courses of action or options by selecting a preferred option or ranking the options from best to worst, which is a process of multi-attribute decision analysis. In daily practice, the use of MADA is essential to signal the best rational choice to the decision maker so that he can allocate limited resources among competing and alternative interests [27].

  1. Basílio M P, Pereira V, Costa H G, et al. A systematic review of the applications of multi-criteria decision aid methods (1977–2022) [J], Electronics, 2022, 11(11): p.1720.”

Point 3: Revise the titles of the tables and format them according to the MDPI author's guide.

Response 3: Thanks for the valuable comment and suggestion. We have checked and modified all titles of the tables according to the MDPI requirements. The revised titles can be found in Tables 1 - 7 in the manuscript.

 Point 4: The conclusion should be more detailed and better address the methods' advantages.

Response 4: This comment is very valuable. We have extensively revised the conclusion section to provide more details of the article and to detail the strengths of the paper's methodology and the value of the paper's research. The specific revisions can be found in the conclusion section of the manuscript.

Reviewer 3 Report

The paper "An improved hesitant fuzzy multi-attribute decision making method for member selection of green supply chain considering green innovation ability" falls within the scope of the journal "Axioms". Still, it doesn't meet the standard quality of the paper that should be published in one prestigious journal in the current version. The paper is interesting, with a well-applied methodology and good structure of the paper. Besides, the paper needs major improvements.

My comments are as follows.

- The title is long and should be more concise. 

- Introduction section is extensive, with some clear explanations.  However, in the introduction section, the following tasks should be fulfilled: the introduction should give an overview of the field's significance and consider the following main questions: What are the gaps in the literature? You have described some tasks from this aspect, but should be improved. What are the main aims of this article?" Also, contributions should be described. 

- Please include the following recently publications:

1) Yazdani, M., Chatterjee, P., & Stević, Ž. (2022). A two-stage integrated model for supplier selection and order allocation: an application in dairy industry. Operational Research in Engineering Sciences: Theory and Applications, 5(3), 210-229.

2) Chen, L. & Su, S. (2022). Optimization of the Trust Propagation on Supply Chain Network Based on Blockchain Plus, J. Intell. Manag. Decis., 1(1), 17-27. https://doi.org/10.56578/jimd010103

3) Puška, A. & Stojanović, I. (2022). Fuzzy Multi-Criteria Analyses on Green Supplier Selection in an Agri-Food Company, J. Intell. Manag. Decis., 1(1), 2-16. 

- Diagram of the research flow should be added and well explained.

- Explanations for the selection of this methodology should be more elaborated.

- Try to show results and discussion in two different sections.

- Future research should be well elaborated.

Author Response

Response to Reviewer 3 Comments:

Point 1: The title is long and should be more concise.

Response 1: Thanks for the valuable comment and suggestion. We have revised the article title to "A green supply chain member selection method considering green innovation capability in a hesitant fuzzy environment".

Point 2: Introduction section is extensive, with some clear explanations.  However, in the introduction section, the following tasks should be fulfilled: the introduction should give an overview of the field's significance and consider the following main questions: What are the gaps in the literature? You have described some tasks from this aspect, but should be improved. What are the main aims of this article?" Also, contributions should be described.

Response 2: We are grateful to your suggestions that improved our manuscript greatly. We have added research contributions in the introduction section and revised the original manuscript. Specifically, the additions are as follows:

“In summary, this paper proposes an improved hesitant fuzzy multi-attribute decision method for member selection in green supply chains, which is applicable to the inter-firm selection decision problem in green supply chains. Also, this paper has the following contributions:

  1. This paper improves the hesitant fuzzy multi-attribute decision-making method considering attribute weight optimization, and provides a more scientific and reasonable attribute weight determination method and a brand-new decision result sorting method for the hesitant fuzzy multi-attribute decision-making problem.
  2. The selection of green supply chain members based on the perspective of green innovation in this paper can promote the innovative research and development of green products by enterprises, and then enhance the green innovation ability of the whole green supply chain, and at the same time provide ideas for future related researches studies and theoretical references for enterprises when selecting green supply chain members.
  3. In this paper, the research on the selection of members in green supply chains is no longer limited to a certain two subjects, but the entire supply chain is studied when the selection decision is made among the enterprises. Therefore, this paper provides a more applicable green supply chain member selection method.”

Point 3: Please include the following recently publications:

1) Yazdani, M., Chatterjee, P., & Stević, Ž. (2022). A two-stage integrated model for supplier selection and order allocation: an application in dairy industry. Operational Research in Engineering Sciences: Theory and Applications, 5(3), 210-229.

2) Chen, L. & Su, S. (2022). Optimization of the Trust Propagation on Supply Chain Network Based on Blockchain Plus, J. Intell. Manag. Decis., 1(1), 17-27. https://doi.org/10.56578/jimd010103

3) Puška, A. & Stojanović, I. (2022). Fuzzy Multi-Criteria Analyses on Green Supplier Selection in an Agri-Food Company, J. Intell. Manag. Decis., 1(1), 2-16.

Response 3: This comment is very valuable. We have added the recommended reference "2)" to the second paragraph of the introduction, "3)" to the fourth paragraph of the introduction, and "1)" to the third paragraph of the literature review. The specific citation forms are as follows:

“8. Chen L, and S. Su. Optimization of the Trust Propagation on Supply Chain Network Based on Blockchain Plus [J]. J. Intelligent Manage. Decision, 2022, 1(1):p. 17-27.

  1. Puška, A., and I. Stojanović. Fuzzy multi-criteria analyses on green supplier selection in an agri-food company. J. Intell. Manag. Decis 2022, 1(1):p. 2-16.
  2. Yazdani, M., Chatterjee, P., and Stević, Ž. (2022), A two-stage integrated model for supplier selection and order allocation: an application in dairy industry, Operational Research in Engineering Sciences: Theory and Applications, 5(3): p.210-229.”

Point 4: Diagram of the research flow should be added and well explained.

Response 4: This comment is very valuable. We have added flowcharts and their explanations in the text .

Point 5: Explanations for the selection of this methodology should be more elaborated.

Response 5: This comment is very valuable. We have added the appropriate content to explain the choice of methods in this paper. The details are as follows:

“Hesitant fuzzy sets, as a kind of fuzzy sets, can well solve the fuzziness and hesitancy of experts at the subjective level. Among them, fuzziness refers to the fact that many times experts cannot give a specific assessment value when evaluating objectives, but can only give a vague and unclear range; hesitancy refers to the fact that when experts evaluate objectives, the assessment value will hesitate between several possible values[11]. With the development of economy and society, realistic decision-making problems are becoming more and more complex, and people often need to evaluate and make decisions on a problem from multiple aspects and dimensions, and the ambiguity and hesitancy in evaluating the problem are becoming stronger. In supply chain chains, to determine which supplier is the best, various fuzzy methods have been used by combining economic and ecological criteria [12]. Therefore, this paper uses hesitant fuzzy sets theory to express decision information, and uses hesitant fuzzy multi-attribute decision-making method to select members of the green supply chain. Therefore, this paper proposes an improved hesitant fuzzy multi-attribute decision making method. This method chooses the hesitation fuzzy theory as its theoretical underpinning and is based on the improved hesitation fuzzy multi-attribute decision making method taking attribute weight optimization into consideration for selection decision. In this method, decision making can fully utilize the decision information provided by each decision maker and give each attribute scientific and reasonable weights while also combining risk-optimism to rank the solutions.”

Point 6: Try to show results and discussion in two different sections.

Response 6: Thank you very much for giving us your opinion. However, the current structure of this type of paper is such that it is a comparative analysis as a complete subsection. It would be indifferent to the analysis of the article if the section were divided into two parts: results and discussion. Our paper is organized with reference to the following papers:

“1. Garg H, Krishankumar R, Ravichandran K S. Decision framework with integrated methods for group decision-making under probabilistic hesitant fuzzy context and unknown weights[J]. Expert Systems with Applications, 2022, 200: 117082. https://doi.org/10.1016/j.eswa.2022.117082

  1. Jiang H, Hu B Q. A decision-theoretic fuzzy rough set in hesitant fuzzy information systems and its application in multi-attribute decision-making[J]. Information Sciences, 2021, 579: 103-127. https://doi.org/10.1016/j.ins.2021.07.094
  2. Zheng Y, Xu Z, He Y, et al. A hesitant fuzzy linguistic bi-objective clustering method for large-scale group decision-making[J]. Expert Systems with Applications, 2021, 168: 114355. https://doi.org/10.1016/j.eswa.2020.114355”

Finally, thank you again for your valuable comments. And we apologize for not being able to make changes according to your comments.

Point 7: Future research should be well elaborated.

Response 7: Thank you very much for your advice. We have elaborated and given more specifics about the future research. The details are as follows:

“Through example analysis, the method of this paper has high practicality in the hesitant fuzzy decision environment of green suppliers, with wide application and closer to practical decision making. In future research, we will consider extending the study of this paper to probabilistic hesitation fuzzy numbers. Although the hesitation fuzzy number can aggregate all the decision values of all decision makers for the same solution, it is only able to aggregate one value when there are decision makers with the same decision value, which ignores the importance of the decision value. However, the probabilistic hesitation fuzzy number can handle such a situation in a probabilistic form. Also, in future research, we will consider applying the method of this paper to some other decision problems, e.g., project site selection, large equipment purchase, etc.”

Round 2

Reviewer 1 Report

The revised article offers several updates to the presentation of the method for the multi-criteria evaluation of suppliers based on the concepts of hesitant fuzzy sets and objective weight.
The authors improved the methodological sections and the description of the application to the case-study. The new diagram of Figure 1 is particularly effective for understanding the method. Some  minor issues in the English form were also solved. Particularly, they have dealt with all questions which I posed to them. I do not agree with some of their conclusions related to the practical effectiveness of the proposed method, but I think this depends on our different backgrounds. Therefore, I think that, apart from some minor flaws in the English form of the newly introduced parts that can be revised by the authors, the paper makes an interesting contribution to the field of MADM evaluation.

Reviewer 2 Report

Dear Authors

I congratulate you on your diligence in implementing the suggestions proposed by the reviewers in the first round of revision. In the current version, I have no improvements to propose.

I wish you success in future research.

Best Regards

Reviewer

Reviewer 3 Report

The paper has been improved according to my suggestions. The authors have provided well reply to my comments. This version of the paper is acceptable.